# CNT and Graphene-Based Transistor Biosensors for Cancer Detection: A Review

**DOI:** 10.3390/biom13071024

**Published:** 2023-06-22

**Authors:** Joydip Sengupta, Chaudhery Mustansar Hussain

**Affiliations:** 1Department of Electronic Science, Jogesh Chandra Chaudhuri College, Kolkata 700033, India; joydipdhruba@gmail.com; 2Department of Chemistry and Environmental Science, New Jersey Institute of Technology, Newark, NJ 07102, USA

**Keywords:** biosensor, cancer, carbon nanomaterials, carbon Nanotube (CNT), graphene, field Effect Transistor (FET)

## Abstract

An essential aspect of successful cancer diagnosis is the identification of malignant tumors during the early stages of development, as this can significantly diminish patient mortality rates and increase their chances of survival. This task is facilitated by cancer biomarkers, which play a crucial role in determining the stage of cancer cells, monitoring their growth, and evaluating the success of treatment. However, conventional cancer detection methods involve several intricate steps, such as time-consuming nucleic acid amplification, target detection, and a complex treatment process that may not be appropriate for rapid screening. Biosensors are emerging as promising diagnostic tools for detecting cancer, and carbon nanotube (CNT)- and graphene-based transistor biosensors have shown great potential due to their unique electrical and mechanical properties. These biosensors have high sensitivity and selectivity, allowing for the rapid detection of cancer biomarkers at low concentrations. This review article discusses recent advances in the development of CNT- and graphene-based transistor biosensors for cancer detection.

## 1. Cancer, Its Severity, and the Necessity of Early Detection

Cancer is a severe and life-threatening condition characterized by the uncontrolled growth and spread of abnormal cells in the body. Its severity arises from the potential to invade and damage surrounding tissues and organs, leading to significant health complications and, if left untreated, even death. Cancer encompasses various forms and severity levels, representing cell growth disorders. The most common type of cancer is carcinoma [1], arising from epithelial cells lining the body’s surfaces. Carcinomas can affect the breast, lung, prostate, colon, and skin, ranging from localized, curable tumors to advanced stages with higher risks and limited treatment options. Sarcomas develop in muscles, tendons, connective tissues, and bones [2]. The severity of sarcomas depends on location, stage, and aggressiveness. Early-stage sarcomas can be cured with surgery, while advanced cases may require radiation and chemotherapy. Leukemias affect blood and marrow [3], disrupting cell formation and function. Severity varies based on type, age, and stage. Treatment options include chemotherapy, radiation, and stem cell transplantation. Lymphomas originate in the lymphatic system [4], either Hodgkin or non-Hodgkin. Severity is determined by subtype, stage, and spread. Treatment includes chemotherapy, radiation, targeted therapies, and stem cell transplantation. Brain, spinal cord, and other CNS cancers differ in severity based on tumor type, location, and stage [5]. Thus, as the severity of cancer depends on the type, stage, and individual factors, management relies on early detection, accurate diagnosis, and effective treatment. Cancer screenings, awareness, and research can contribute to early detection and successful therapy.

The early detection of cancer is of utmost importance due to several key reasons [6] (Figure 1). Firstly, early-stage cancer is often more treatable and has a higher likelihood of successful outcomes. Detecting cancer at an early stage allows for a wider range of treatment options [7], including less invasive procedures, leading to a higher chance of complete remission. Secondly, early detection can help prevent the progression of cancer to advanced stages. When cancer is detected late, it may have already spread to other parts of the body (metastasis) [8], making it more challenging to treat and control. Early detection increases the chances of localized treatment, reducing the risk of metastasis and improving overall prognosis. Furthermore, early detection can significantly impact patient survival rates [9]. Many cancers, when identified early, have higher long-term survival rates compared to cases where the disease is diagnosed at a later stage. Regular screenings and early diagnostic tests can lead to timely interventions, improving patient outcomes and extending their life expectancy. Additionally, the early detection of cancer can reduce the physical, emotional, and financial burden on individuals and healthcare systems [10]. Early-stage treatments often require fewer resources and interventions, resulting in lower healthcare costs and a reduced impact on the patient’s quality of life. Thus, the severity of cancer and the necessity for its early detection stems from a desire to avoid the dangerous potential of advanced-stage cancer, reduced treatment options, and poorer prognosis. Early detection plays a vital role in improving treatment success rates, preventing metastasis, increasing survival rates, and reducing the overall burden associated with cancer on individuals and healthcare systems.

## 2. Cancer Detection and Biosensors

Cancer detection techniques play a critical role in the early diagnosis and treatment of cancer. Various methods are employed to identify the presence of cancer cells or cancer-related abnormalities in the body. These techniques include imaging techniques such as X-rays [11], computed tomography (CT) [12], magnetic resonance imaging (MRI) [13], mammography [14], pap smear [15], colonoscopy [16], prostate-specific antigen (PSA) test [17], and ultrasounds [18], all of which provide detailed images of the body to identify tumors or abnormal growths. A biopsy [19] is another important technique, wherein a sample of tissue or cells is taken for examination under a microscope to determine if cancer is present. Additionally, molecular techniques like polymerase chain reaction (PCR) [20] and next-generation sequencing (NGS) [21] are used to analyze genetic material and identify specific mutations associated with cancer. Blood tests and biomarker analysis are also utilized to detect specific proteins or substances in the blood that may indicate the presence of cancer. Nevertheless, these conventional techniques have their own advantages and disadvantages. X-rays offer the advantages of being non-invasive, affordable, and widely accessible; however, they have limitations in their ability to detect small tumors and provide detailed information. CT scans, on the other hand, provide highly detailed images and are effective in detecting small lesions and identifying tumors, but they come with the drawback of radiation exposure and potentially higher costs. MRI stands out for its ability to provide excellent anatomical detail without radiation, although it can be costly and time-consuming. Mammography is a valuable tool for effectively detecting early-stage breast cancer due to its wide availability and relative affordability, yet it can yield false-positive results and present challenges with dense breast tissue. Pap smears are relatively low-cost and widely used for detecting cervical precancerous changes and early-stage cancer, but they carry the risk of false-negative results, necessitating regular screenings. Colonoscopies, recognized as the gold standard for colon cancer screening, are capable of both detecting and removing polyps, thereby reducing the risk of colorectal cancer; however, this procedure is invasive, requires preparation, and carries potential risks. The PSA test, a simple blood test for prostate cancer, detects early-stage cancer, but it is prone to false-positive results and lacks the ability to differentiate between aggressive and indolent cancers. Ultrasounds, a non-invasive and radiation-free option, are relatively cost-effective and widely used for various cancers, although they have limitations in terms of tissue penetration and detailed anatomical information. Biopsies serve as a definitive diagnostic tool to help determine the type and stage of cancer; however, a biopsy is an invasive procedure associated with certain risks. PCR, with its ability to amplify and detect specific DNA sequences, is highly sensitive and useful for detecting mutations and infectious agents linked to cancer; nevertheless, it requires specialized equipment and expertise. NGS, a sophisticated DNA sequencing method that can analyze multiple genes simultaneously, plays a valuable role in identifying genetic mutations associated with cancer and guiding personalized treatment; however, it is a complex and expensive technique that demands bioinformatics expertise for data analysis.

Furthermore, advanced technologies like biosensors [22] and liquid biopsies [23] are emerging as promising methods for the non-invasive and early detection of cancer. Liquid biopsies are innovative diagnostic tests that involve the analysis of various components—such as circulating tumor cells (CTCs), cell-free DNA (cfDNA), and extracellular vesicles—present in a patient’s blood or other bodily fluids. They offer a non-invasive approach to detecting and monitoring cancer. Liquid biopsies contribute to early cancer detection by providing valuable information about the presence of cancer-related genetic mutations, alterations, and biomarkers. These tests can help identify circulating tumor cells or fragments of tumor DNA shed into the bloodstream, even at early stages when traditional imaging techniques may not detect the tumor. Liquid biopsies are particularly beneficial for monitoring treatment response, assessing minimal residual disease, and detecting tumor recurrence. By enabling early detection, liquid biopsies have the potential to improve patient outcomes through timely intervention, personalized treatment selection, and the monitoring of treatment efficacy. The early detection of cancer is a critical factor in improving patient outcomes, and emerging technologies such as Photonic Crystal Enhanced Fluorescence (PCEF) [24] and Photonic Resonator Absorption Microscopy (PRAM) [25] hold great promise in this endeavor. PCEF leverages the unique properties of photonic crystals to enhance fluorescence signals from cancer-specific biomarkers, enabling the sensitive and specific detection of cancer at an early stage. By carefully engineering the photonic crystal structures, PCEF minimizes background fluorescence and unwanted scattering, resulting in an improved signal-to-noise ratio and enhanced sensitivity. This technology has the potential to revolutionize cancer diagnostics by enabling the detection of low-abundance biomarkers and facilitating multiplexed analysis of multiple cancer-related molecules simultaneously. On the other hand, PRAM utilizes high-Q photonic resonators to enhance the absorption of light by cancer cells or tissues. By precisely tuning the resonant frequencies of the resonators, PRAM can selectively amplify the absorption of specific wavelengths, thereby enhancing the contrast between cancerous and healthy cells. This label-free imaging technique provides detailed information about cellular morphology and molecular composition, aiding in the early identification of cancerous changes. The integration of PCEF and PRAM into clinical practice has the potential to revolutionize cancer screening and improve survival rates through early detection and intervention. 

The rationale for exploring the combination of plasmonic materials with low-dimensional substrates for cancer detection stems from the unique properties and advantages offered by this synergistic approach. Plasmonic materials, such as metal nanoparticles, exhibit strong interactions with light at the nanoscale, resulting in enhanced electromagnetic fields and localized surface plasmon resonances. These properties enable the highly sensitive detection of biological analytes, including cancer biomarkers, through surface-enhanced Raman scattering (SERS) [26], localized surface plasmon resonance (LSPR) [27], or plasmon-enhanced fluorescence [28]. By integrating plasmonic materials with low-dimensional substrates, such as graphene, two-dimensional transition metal dichalcogenides (TMDs) [29], or nanowires, the advantages are further enhanced. The integration of plasmonic materials with low-dimensional substrates enables improved sensitivity, efficient signal transduction, and versatile platform design.

The continuous development and improvement of these detection techniques are vital in enabling early diagnosis, personalized treatment, and better patient outcomes in the fight against cancer.

It is important to note that no single detection technique is perfect, and a combination of different methods may be used to achieve accurate cancer diagnosis. Advancements in technology, such as the development of novel biomarkers, improved imaging techniques, and innovative diagnostic tools, continue to enhance cancer detection and improve patient outcomes. Regular cancer screenings and early detection remain crucial for timely interventions and effective treatment.

Biosensors have shown great potential for the early detection of cancer due to their high sensitivity and specificity. One type of biosensor commonly used for this purpose is the protein-based biosensor [30]. These biosensors utilize specific proteins or antibodies that can recognize and bind to cancer-specific biomarkers present in the body. By detecting the presence and concentration of these biomarkers, biosensors can identify the early stages of cancer before symptoms become apparent. For example, Teker et al. [31] and Kim et al. [32] employed CNT-based technology, while Majd et al. [33] and Kim et al. [34] utilized graphene-based protein biosensors for the purpose of early cancer detection. Another type of biosensor used for early cancer detection is the nucleic acid-based biosensor [35]. These biosensors can detect specific DNA or RNA sequences that are associated with cancer cells. By analyzing the genetic material in body fluids or tissues, nucleic acid biosensors can identify genetic mutations or alterations that indicate the presence of cancer. In the pursuit of early cancer detection, Ramnani et al. [36] and Li et al. [37] employed CNT-based technology, whereas Cai et al. [38] and Gao et al. [39] utilized biosensors based on graphene for the detection of nucleic acids associated with cancer.

Moreover, biosensors integrated with microfluidic systems offer advantages for early cancer detection. Microfluidic biosensors [40] combine microfluidics and biological sensing elements to detect and analyze specific molecules or biomarkers in fluid samples. These devices consist of microchannels, chambers, and functionalized surfaces that interact with the target analyte. The precise manipulation and control of fluids at the microscale enable efficient sample handling and analysis. Microfluidic biosensors offer enhanced sensitivity by optimizing mass transfer, reducing reagent consumption, and facilitating rapid reaction kinetics through their small size and design. The miniaturized channels and functionalized surfaces provide increased surface area-to-volume ratios, promoting higher binding efficiency between the target analytes and the sensing elements, resulting in improved sensitivity and detection limits. Furthermore, microfluidic biosensors allow for multiplexed detection, enabling simultaneous analyses of multiple analytes in a single assay. By incorporating multiple sensing elements or channels, different analytes can be detected and analyzed concurrently, leading to time and cost efficiency by eliminating the need for separate assays and enabling comprehensive analysis in a single experiment. The development of biosensors for the early detection of cancer is driven by the need for non-invasive and rapid diagnostic tools. Biosensors have the potential to revolutionize cancer screening programs by offering simple and cost-effective methods for early detection [41]. With further advancements in biosensor technology and ongoing research, biosensors hold promise in improving cancer survival rates through early diagnosis and timely intervention (Figure 2).

The motivation for this paper stems from the urgent need for improved cancer detection methods that can enable early-stage diagnosis and personalized treatment. Conventional diagnostic approaches often face limitations in terms of sensitivity, selectivity, and real-time monitoring. By harnessing the power of CNT- and graphene-based transistor biosensors, this paper aims to shed light on their potential to overcome these limitations and propel cancer diagnostics to new heights. The nanotechnology perspective drives the exploration of CNT- and graphene-based biosensors as they offer unparalleled advantages for cancer detection. The ability to directly interface with biological molecules and detect minute concentrations of cancer biomarkers with high sensitivity and selectivity is a game-changer in the fight against cancer. By leveraging the unique properties of CNTs and graphene, nanotechnology strives to unlock novel avenues for early detection, accurate diagnosis, and personalized treatment strategies.

## 3. Carbon Nanomaterial-Based Transistor Biosensors for Cancer Detection

Carbon nanomaterial-based field-effect transistor (CNFET) biosensors have emerged and shown promise in the early detection of cancer [43]. These biosensors mostly utilize CNTs or graphene as the active sensing component, offering unique properties that enhance the sensitivity and performance of the biosensors. The sensing mechanism of CNTs and graphene involves their unique electrical properties, which are sensitive to changes in the surrounding environment. In the case of CNTs, their exceptional electrical conductivity allows them to act as conductive channels in a sensing device. When CNTs are functionalized with specific biomolecules or receptors, such as antibodies or aptamers, they can selectively bind to target analytes, such as cancer biomarkers [44]. The binding event causes a change in the electrical conductivity of the CNTs, either due to charge transfer or changes in the charge distribution near the CNT surface. This change in conductivity can be measured and correlated to the presence and concentration of the target analyte, enabling label-free detection [45]. On the other hand, graphene has high electrical conductivity and a large surface area, which makes it an excellent candidate for sensing applications. In graphene-based sensors, the adsorption of target analytes onto the graphene surface alters the local charge distribution and doping level of the material [46]. This change in charge distribution affects the electrical conductivity of graphene, which can be detected and quantified. Graphene sensors can operate in various modes, such as FET configuration, where the gate voltage is modulated by the presence of analytes, or in resistive mode, where the change in resistance of graphene is directly measured.

Regarding CNT-based FET biosensors [47] (Figure 3), single-walled CNTs (SWCNTs) or multi-walled CNTs (MWCNTs) are used as the conducting channel. CNTs offer significant advantages in terms of their electrical properties and sensitivity. With their excellent electrical conductivity, CNTs efficiently transport charge carriers, such as electrons, along their one-dimensional structure, making them highly desirable for electronics and sensor applications. CNTs also exhibit exceptional sensitivity to changes in temperature, pressure, or the presence of specific molecules, enabling them to detect and quantify small quantities of substances with precision. Their high aspect ratio and large surface area enhance interaction with target molecules, resulting in improved sensitivity for sensing applications. The functionalization of CNTs with specific bioreceptors, such as antibodies or aptamers, enables the selective recognition and binding of cancer biomarkers. The CNT functionalization with specific bioreceptors involves modifying the CNT surface to enable the selective binding and recognition of target molecules. This process includes preparing the CNTs through purification and functionalization, followed by attaching linker molecules that provide a stable interface. These linkers are coated with biomolecules to enhance binding, and the desired bioreceptor molecules are then immobilized onto the functionalized CNTs through techniques like covalent bonding or physical adsorption. The choice of bioreceptor depends on the target molecule, with antibodies offering high specificity and affinity for specific antigens, while aptamers provide synthetic nucleic acid sequences capable of binding to a wide range of targets with high specificity. Finally, the binding event induces a change in the conductance of the CNT-FET, which can be measured and correlated with the presence and concentration of the biomarker. This label-free detection approach allows for the sensitive and real-time monitoring of cancer biomarkers, enabling early detection.

On the other hand, Graphene-based FET biosensors [48] (Figure 3) utilize graphene as the conducting channel. Graphene’s unique electrical and mechanical properties make it an excellent platform for biosensing. By functionalizing graphene with biomolecules, such as DNA probes or antibodies, the biosensor can selectively capture cancer-specific biomarkers. The interaction between the biomarker and the functionalized graphene surface causes a change in the electrical conductance, which can be measured and quantified. Graphene-based FET biosensors offer high sensitivity, rapid response, and the potential for multiplexed detection of multiple biomarkers.

Overall, CNFET biosensors hold significant promise for early cancer detection, offering the potential to improve patient outcomes through timely diagnosis, personalized treatment, and improved survival rates. The incorporation of CNTs or graphene in FET biosensors offers several advantages for cancer detection. These include high sensitivity, rapid response, label-free detection, and the potential for multiplexed analysis. CNTs and graphene provide a large surface area for biomarker binding, enhancing the sensitivity of the biosensor. Moreover, their unique electrical properties allow for the real-time monitoring and quantification of cancer biomarkers with high precision. Furthermore, the integration of CNTs or graphene with FET biosensors enables the development of miniaturized and portable microfluidic devices [49], facilitating point-of-care testing and decentralized cancer screening. These biosensors enable the detection of cancer biomarkers at low concentrations, facilitating early diagnosis and intervention. Their sensitivity and specificity help reduce false-positive results and improve diagnostic accuracy [50].

### 3.1. CNT-Based Transistor Biosensors for Cancer Detection

Teker et al. [31] developed a novel method for detecting breast cancer cells in real-time utilizing CNT-FET-based label-free protein biosensors linked with antibodies. Specific and nonspecific monoclonal antibodies (mAbs), as well as the insulin-like growth factor 1 receptor (IGF1R), were able to significantly decrease the CNT-FET devices’ conductivity. Upon the introduction of human BT474 and MCF7 breast cancer cells, an increase in conductance was observed exclusively in the IGF1R-specific mAb-CNT-FET devices. This increase was significantly associated with the number of IGF1Rs found on the breast cancer cell membranes. Moreover, the device conductivity was found to be much higher in the presence of MCF7 breast cancer cells than in BT474 cells, possibly due to a higher number of IGF1 receptors present. Conversely, the nonspecific mAb-CNT-FETs exhibited no change in conductance when exposed to BT474 and MCF7 breast cancer cells. These findings strongly indicate that the specific binding between the IGF1 receptor and its corresponding mAb induces a remarkable alteration in the electrical conductivity of CNT-FET devices. Subsequently, mAb-CNT-FETs exhibit great potential as biosensors for the detection of circulating cancer cells in blood samples.

Kim et al. [32] presented a simple and extremely sensitive technique for the real-time monitoring of the PSA-ACT complex, a prostate cancer marker. The approach employed label-free protein biosensors based on a CNT-FET. The researchers employed 1-pyrenebutanoic acid succinimidyl ester (PASE) as linkers and 1-pyrenbutanol as spacers in the CNT-FET biosensor. For optimal performance, the CNT-FETs underwent functionalization using five distinct solutions with varying ratios of linkers to spacers. The binding event between the target PSA-ACT complex and the receptor was monitored by observing the gating effect arising from charge alterations in the target PSA-ACT complex. The results showed that CNT-FET biosensors that were only altered with linkers could not find target proteins unless a PSA-ACT complex solution with a very high concentration (about 500 ng/mL) was added. However, when modified with a linker-to-spacer ratio of 1:3, the biosensors exhibited the capability to detect 1.0 ng/mL of the target protein without any pretreatment. Moreover, the CNT-FET, modified with both linkers and spacers, effectively blocked non-target proteins, allowing for the selective detection of the target protein in human serum.

In their research, Jones et al. [51] conducted a comprehensive investigation to assess the feasibility of utilizing a CNT-FET as a diagnostic tool for quantifying cancer biomarkers in serum. Specifically, they focused on measuring insulin-like growth factor-1 (IGF-1) in a mouse model of Breast Cancer Susceptibility 1-related breast cancer. The selection of IGF-1 was motivated by its high relevance in breast cancer and the challenges associated with conventional measurement methods due to specific IGF-1 serum-binding proteins. The study aimed to compare the performance of radioimmunoassay and CNT-FET assays in measuring serum IGF-1 levels in this preclinical model of human breast cancer. The findings demonstrated a significant correlation between the two platforms for the detection of serum IGF-1. Notably, the CNT-FETs exhibited several advantages over the radioimmunoassay, including the requirement of only one antibody, real-time results, and a reduced volume of mouse serum by approximately 100-fold. The CNT-FET assay also proved to be facile and more rapid than the radioimmunoassay. Moreover, the low serum sample requirement of the CNT-FETs holds great potential, particularly in studies with limited access to human clinical samples.

Lerner et al. [52] presented a pioneering approach to detect osteopontin (OPN), an emerging biomarker for prostate cancer. They devised a method by coupling a genetically engineered single-chain variable fragment (scFv) protein known for its high affinity to OPN with a CNT-FET. The scFv was covalently attached to the CNT-FETs through chemical functionalization with diazonium ions. Importantly, this functionalization process maintained the biological binding site’s activity for OPN. Electron transport measurements demonstrated that the modified CNT-FET exhibited the potential to detect OPN binding with the complementary scFv protein. Notably, the drain current exhibited a concentration-dependent increase within the clinically relevant range, with an impressive detection limit of approximately 30 femtomolar (fM). Moreover, the experiments demonstrated an antigen-specific response proportionate to the concentration, spanning a broad range from 1 picogram per milliliter (pg/mL) to 1 microgram per milliliter (μg/mL). These observations strongly support the notion that changes in scattering at the sites of scFv protein-occupied defects on the sidewalls of CNTs underlie the detection mechanism.

Small RNA molecules, ranging in size from 19 to 23 nucleotides, perform vital functions in the regulation of genes, embryonic development, hematopoiesis, and various forms of cancer. In an innovative study, Ramnani et al. [36] presented an extraordinary, highly specific, label-free, and rapid electronic method for detecting microRNAs (miRNAs). This accomplishment was made by employing a CNT-FET that had been functionalized with the p19 protein derived from the Carnation Italian ringspot virus. Their study considered miRNA-122a as the target molecule, which was initially combined with a PASE probe. Subsequently, the resulting probe-miRNA duplex was assessed by measuring changes in resistance with respect to the biosensor caused by its interaction with p19. Remarkably, p19 exhibited a size-dependent but sequence-independent preference for 21 to 23 base pair RNA duplexes. The biosensor exhibited an impressive dynamic range, extending up to concentrations as low as 10 to the power of −14 M, capable of detecting miRNA concentrations as minimal as 1 attomolar (aM). These findings pave the way for the development of simple, affordable, point-of-care techniques for the early detection of miRNA biomarkers in the diagnosis of various diseases, including cancer.

Prostate-specific antigen (PSA) is an enzyme glycoprotein responsible for the breakdown of ejaculate and cervical mucus. While it exists in small quantities in the serum of males with healthy prostates, elevated levels of this enzyme often indicate the presence of prostate cancer or other prostate disorders. In an innovative proposal, a bio-sensing platform based on a horizontally aligned SWCNT-FET was introduced by Chen et al. [53] to enable real-time and highly sensitive protein detection. The fabrication process involved synthesizing aligned nanotubes on a quartz substrate using CVD via catalyst (Cobalt(II) acetate tetrahydrate) contact stamping. The efficacy of FET biosensors in detecting prostate-specific antigens (PSA) in real-time has been successfully demonstrated. As the concentration of PSA increased, the discharge current of the biosensor decreased exponentially, as measured by kinetic methods. This finding indicates that the proposed FET sensor can quantitatively detect proteins within a detection range of up to 1 µM. Impressively, the proposed platform achieved a limit of detection (LOD) of 84 pM, highlighting its remarkable sensitivity and accuracy.

Interleukin-6 (IL-6), a prominent cytokine found within the tumor microenvironment, exhibits dysregulation and high concentration in cancer. Its overexpression spans diverse tumor types, underscoring the robust connection between inflammation and cancer. Within the tumor microenvironment, heightened IL-6 levels play a pivotal role in driving tumorigenesis by modulating numerous signaling pathways and governing critical aspects of cancer progression. Khosravi et al. [54] demonstrated a rapid and label-free method for detecting Interleukin-6 (IL-6) by employing CNT microarrays with aptamers as the molecular recognition elements. The CNT-FET biosensors were functionalized with IL-6 aptamers that were conjugated with PASE. When the aptamer-functionalized nanotube surface was exposed to IL-6, the conductance decreased in proportion to the concentration of IL-6. The presence of IL-6 in the range of 1 pg/mL to 10 ng/mL was successfully detected in both buffer and blood samples using IL-6 aptamers conjugated with PASE as the molecular recognition element. Consequently, the nanotube microarray exhibited the capability to capture cancer cells through antibody binding and detect IL-6 using aptamers.

A similar study by Chen et al. [55] presented a highly sensitive, easy-to-use assay without the need for labeling, showcasing the detection of IL-6 using liquid-gated FET sensors. These sensors were constructed based on horizontally aligned SWCNTs that were grown on a quartz substrate using CVD. The method involved observing the drain current of the transistor as IL-6 bound to its receptor antibody (IL-6R) immobilized on SWCNTs. In this study, the device demonstrated exceptional sensitivity (LOD = 1.37 pg/mL) due to the minimized tube-to-tube contact resistance. It also exhibited excellent selectivity, which resulted from the specific interaction between IL-6R and IL-6. The strong adherence of CNT to the quartz substrate and exact horizontal alignment of the nanotubes contributed to the device’s extraordinary stability. The proposed immunosensor holds great promise for the early detection of various diseases, as evidenced by the fluctuating levels of IL-6 in the bloodstream.

Tumor-derived miRNAs present in exosomes might play crucial roles in the initiation and advancement of cancer, making them promising biomarkers for early detection and prognosis monitoring. Li et al. [37] developed a highly sensitive, label-free, and stable biosensor utilizing a semiconducting CNT film sorted by a polymer introduced for the detection of exosomal miRNAs (Figure 4). The biosensor, based on a FET with a floating-gate structure, incorporated an ultrathin Y_2_O_3_ layer as an insulator and utilized gold nanoparticles (AuNPs) on Y_2_O_3_ as linkers to immobilize probe molecules. Attaching a thiolated oligonucleotide probe to the AuNP-coated sensing area and monitoring variations in the current before and after hybridization of the immobilized DNA probe with the target miRNA enabled the biosensor to detect miRNA21. This approach demonstrated exceptional sensitivity (limit of detection: 0.87 aM) and specificity. Moreover, the FET biosensor successfully distinguishes between healthy individuals and breast cancer patients when applied to clinical plasma samples. Additionally, the sensor holds potential for integration with microfluidic technology, enabling the rapid and highly sensitive detection of multiple tumor biomarkers on a single chip, thus facilitating precise tumor diagnosis.

The proof of concept for detecting the AKT2 gene, which serves as a typical biomarker for metastatic and primary triple-negative breast cancer (TNBC), has been achieved using a highly advanced approach by Ma et al. [56]. Specifically, the integration of a cutting-edge all-CNT thin-film transistor platform with tetrahedral DNA nanostructures (TDNs) has enabled the development of a label-free detection system that is both highly sensitive and highly selective for the AKT2 gene associated with triple-negative breast cancer. Through a systematic analysis of the sensing measurements obtained from the all-CNT TFT biosensors, a detailed investigation into the underlying sensing mechanisms was conducted. By incorporating TDNs, the all-CNT TFT biosensors exhibited significantly enhanced response rates, ranging from a minimum of 35% to an impressive maximum of 98%, surpassing the performance of conventional single-stranded DNA (ssDNA) probes due to the superior efficiency of DNA hybridization. Furthermore, the impact of TDNs’ linker length on the biosensor’s performance was demonstrated, highlighting the 4T-linker TDNs as the most effective in generating the highest response from the all-CNT TFT biosensors. The biosensors successfully detected varying concentrations of DNA by utilizing TDNs with a 4T-linker, achieving a wide linear detection range spanning from 1 pM to 1 μM, with an estimated limit of detection (LOD) of 2 fM. The biosensors exhibited exceptional selectivity towards non-complementary DNA targets and demonstrated favorable repeatability.

The early detection of lung cancer is of utmost importance, and carcinoembryonic antigen (CEA) serves as a well-established biomarker for this purpose. However, the full clinical potential of CEA has yet to be realized due to the demanding requirements for high sensitivity and broad detection capabilities. To address this challenge, Li et al. [57] developed an innovative biosensing approach using a floating gate FET combined with a semiconducting CNT film and an undulating yttrium oxide (Y_2_O_3_) dielectric layer as the biosensing interface (Figure 5). The introduction of the rippling biosensing interface in the proposed device significantly expanded the detection range, enabling detection from ultra-low concentrations of 1 fg/mL to 1 ng/mL. Furthermore, the sensitivity and detection limit were optimized, achieving an impressive 72 ag/mL, attributed to the increased binding sites for the detection probes on the sensing interface and enhanced electric double-layer capacitance. Importantly, the developed sensing platform demonstrated its robust functionality in the presence of complex fetal bovine serum, thus exhibiting great potential for the early screening of lung cancer.

The prior discussion of transistor biosensors based on CNT for detecting cancer biomarkers is summarized in Table 1.

### 3.2. Graphene-Based Transistor Biosensors for Cancer Detection

Prostate-specific antigen/a1-antichymotrypsin (PSA-ACT) complex detection using a reduced graphene oxide (rGO) field-effect transistor (rGO-FET) biosensor was reported by Kim et al. [58]. In this device, the rGO channel was formed through the reduction of graphene oxide (GO) nanosheets, which were interconnected via a self-assembly process. The interaction between PSA-ACT complexes and PSA monoclonal antibodies on the surface of the rGO channel triggered a linear response in the shift of the gate voltage. The rGO-FET demonstrated an extraordinary capability to detect protein–protein interactions at femtomolar levels, exhibiting an impressive dynamic range spanning over six orders of magnitude. This biosensor achieved an unprecedented limit of detection as low as 1.1 fM in a 1 µM buffer solution. The exceptional ultrasensitivity, characterized by the low limit of detection in the femtomolar range and the substantial dynamic range, can be attributed to the densely immobilized receptor biomolecules and the virtually non-existent non-specific binding.

A cost-effective and straightforward approach called low-cost and facile shrink lithography was utilized by Zhang et al. [59] to employ suspended graphene nanoribbons (GNRs) in the development of an ion-sensitive FET (ISFET) suitable for cancer marker sensing applications. By integrating shrink thermoplastic film and a hot embossing process, 50 nm wide GNR patterns were achieved in an inexpensive and uncomplicated manner. Following annealing, the suspended GNR ISFET exhibited ambipolar characteristics, indicating an amplified ambipolar effect that showcased a larger intrinsic bandgap compared to non-annealed GNRs. In the realm of cancer biomarker sensing, the functionalized suspended GNRs equipped with specific anti-PSA antibodies as bio-receptors displayed exceptional capabilities in detecting prostate-specific antigens with an impressive sensitivity of 0.4 pg/mL. To provide a comprehensive evaluation, the performance of unsuspended GNRs and microscale graphene sheet-based ISFETs (designed and fabricated using the same methods and subjected to identical measurement conditions) was compared. The results unequivocally demonstrated that the suspended GNR ISFET surpassed its counterparts in terms of sensitivity and detection limit. The use of suspended GNRs in the ISFET biosensor offers advantages of reduced electrical noise and enhanced charge carrier mobility. The suspended configuration minimizes substrate-induced interference, improving the signal-to-noise ratio and boosting sensitivity and detection limits. Additionally, the free-standing GNRs exhibit superior charge transport properties by minimizing scattering effects, resulting in enhanced sensing performance and lower detection limits.

Ongoing research has revealed a rising body of evidence suggesting the potential of miRNAs as valuable biomarkers for diagnosing and prognosing human cancer, as well as for exploring therapeutic targets and tools. Cai et al. [38] presented an innovative biosensor utilizing a hybrid configuration of gold nanoparticles (AuNPs) and an rGO-FET for the exceptional, label-free, and highly sensitive detection of microRNA (miRNA). The fabrication process involved the drop-casting of a suspension containing rGO onto the surface of the sensor, followed by the precise decoration of AuNPs onto the rGO surface. To enable the detection of miRNA, a specific peptide nucleic acid (PNA) probe was strategically immobilized on the AuNPs surface. The detection mechanism relied on the precise hybridization of PNA with the target miRNA. The developed FET biosensor achieved an astonishingly low detection limit of 10 femtomolar (fM), showcasing its exceptional sensitivity. Furthermore, the biosensor demonstrated notable accuracy in distinguishing between complementary miRNA, miRNA with a one-base mismatch, and noncomplementary miRNA. This attribute further enhanced its potential as an effective diagnostic method for gene-related diseases. Additionally, researchers successfully employed this highly sensitive and selective assay to detect miRNA in serum samples, thereby expanding its applicability and confirming its potential as a valuable tool in medical diagnosis.

Cumulative scientific evidence underscores the significance of aberrant MMP-7 expression in the initiation and advancement of tumors. Enhanced MMP-7 protein levels have been consistently observed in various human cancer cell lines and tissues spanning a wide range of malignancies such as esophageal, ovarian, breast, lung, and other tumor types. A novel and rapid method for the accurate detection of MMP-7 was presented by Chen et al. [60], who utilized a polypeptide (JR2EC) functionalized rGO-FET platform. The method exhibited a short lag time of 5 min and exceptional sensitivity, with a limit of detection of 10 ng/mL in the buffer. The enhanced sensitivity observed can be ascribed to the substantial decrease in the overall charge of the polypeptides following MMP-7 digestion. Importantly, the proposed assay successfully detected MMP-7 in human plasma, achieving a limit of detection of 40 ng/mL. Additionally, this assay allowed for the simultaneous assessment of both MMP-7 activity, and concentration and high specificity was demonstrated by effectively distinguishing MMP-1, a protease within the same family. Thus, the suggested assay holds great promise for cancer diagnosis and tumor identification.

Rajesh et al. [61] successfully prepared high-quality graphene through chemical vapor deposition. This graphene was then transferred onto gold electrodes that were microfabricated on a SiO_2_/Si wafer, resulting in an array consisting of 52 graphene-based FET (GFET) devices. To enhance its properties, platinum nanoparticles (PtNPs) were introduced to create a hybrid nanostructure. In addition, the scientists implemented a scalable manufacturing procedure for biosensor arrays by utilizing PtNP/graphene devices that were functionalized with genetically engineered thiol-containing scFv antibodies specifically designed to recognize the breast cancer biomarker protein HER3. Remarkably, the device displayed a concentration-dependent response within a broad concentration range of HER3 (from 300 fg/mL to 300 ng/mL) when tested in phosphate-buffered saline, with a limit of detection of 300 fg/mL.

A label-free graphene immunosensor for carcinoembryonic antigen (CEA) was presented by Zhou et al. [62] based on non-covalent modification with antibody molecules (Figure 6). The immobilization of anti-CEA antibodies onto the graphene surface was achieved through non-covalent interactions. Graphene interacted non-covalently with a bifunctional molecule, PASE, which contained a pyrene and a reactive succinimide ester group. The abundantly present primary and secondary amines on the surface of anti-CEA facilitated the reaction with the succinimidyl ester groups, enabling the immobilization of anti-CEA antibodies on single-layer graphene. The resulting immunosensor, modified with anti-CEA, effectively monitored the real-time interaction between CEA protein and anti-CEA with high specificity, allowing for the selective electrical detection of CEA. The limit of detection (LOD) for CEA was less than 100 pg/mL.

Haslam et al. [63] employed chemical vapor deposition (CVD) to grow graphene and utilized it to develop highly sensitive immunosensors known as label-free graphene FETs (GFETs) for the detection of Human Chorionic Gonadotropin (hCG), a glycoprotein biomarker associated with certain types of cancer. The GFET sensors were meticulously fabricated on a Si/SiO_2_ substrate using photolithography, wherein chromium was evaporated, and gold contacts were sputtered onto the substrate. To enhance the performance of the GFET channels, a linker molecule PASE modified with NHS chemistry was introduced to facilitate the binding of an immobile anti-hCG monoclonal antibody onto the graphene surface. By subjecting the antibody to different concentrations of the hCG antigen, the researchers demonstrated that the GFET sensors exhibited an impressive limit of detection, measuring below 1 pg/mL. Furthermore, the study highlighted the impact of annealing on the performance of the GFET sensors, revealing that this additional step could substantially enhance their functionality and potentially lead to even greater sensitivity in the detection of hCG.

Majd et al. [33] coated the flexible poly-methyl methacrylate (PMMA) substrate with a large surface area nanosheets of rGO for the fabrication of an n-type FET platform. This platform was employed as a signal transducer in a liquid-ion-gated FET-type biosensor designed to detect the cancer marker protein CA 125. To enable the attachment of anti-CA125 ssDNA onto the rGO, a sequential chemical process was employed. Initially, the ssDNA was attached to carboxylated MWCNTs using N-(3-dimethylaminopropyl)-N’-ethylcarbodiimide hydrochloride and N-hydroxysuccinimide chemistry. Subsequently, the CA125 ssDNA molecules, wrapped around the MWCNTs, were stacked onto the surface of the graphene through p–p interaction. By integrating the CA125 ssDNA-wrapped MWCNTs onto the rGO platform, the researchers achieved the highly sensitive and selective detection of CA125 in human serum samples. The biosensor demonstrated a linear dynamic range spanning from 1.0 × 10^−9^ to 1.0 U/mL, with an extremely low limit of detection of 5.0 × 10^−10^ U/mL.

Kim et al. [34] developed a biosensor to detect alpha-fetoprotein (AFP) in plasma, an essential marker for diagnosing hepatocellular carcinoma (HCC) in humans. AFP detection was carried out using a GFET in HCC patient plasma and a phosphate-buffered saline (PBS) solution. To immobilize an anti-AFP antibody, the GFET was functionalized with 1-pyrenebutyric acid N-hydroxysuccinimide ester (PBASE). The voltage shift of the Dirac point was assessed to detect AFP binding to the immobilized anti-AFP on the GFET channel surface. The anti-AFP-immobilized GFET biosensor successfully detected AFP at a concentration of 0.1 ng/mL in PBS, achieving a detection sensitivity of 16.91 mV. In the plasma of HCC patients, the biosensor detected AFP at a concentration of 12.9 ng/mL, at a detection sensitivity of 5.68 mV. The sensitivity of detection varied depending on the AFP concentration in both PBS and HCC patient plasma.

A solution-gated GFET was developed by Kim et al. [64] for the detection of TP53 DNA. The TP53 DNA, a gene associated with cancer, was applied to the graphene active layers to evaluate the efficiency of the solution-gated GFET as a sensor. Following the immobilization of the probe DNA, the researchers hybridized the target DNA at various concentrations (ranging from 10 μM to 1 nM). The achieved limit of detection was 1 nM. The shift of the Dirac point in the GFET was observed due to the doping effect induced by the negative charges of the DNA present on the graphene surface.

Intracellular glucose metabolism encompasses two key pathways: aerobic respiration and glycolysis. Typically, normal cells employ glucose oxidative phosphorylation as their primary metabolic mechanism. However, in the context of cell carcinogenesis, the process deviates from aerobic metabolism and adopts glycolysis as its energy source. This metabolic alteration leads to the accumulation of lactic acid in the vicinity of the tumor tissue, ultimately resulting in the creation of an acidic microenvironment. Consequently, the expeditious and highly responsive detection of lactic acid within biological systems holds the potential to facilitate the routine examination and identification of tumor cells. Through the in situ pyrolysis of cerium nitrate-pretreated Jerusalem artichoke stalk, nanoceria-loaded porous carbon was synthesized by Bi et al. [65]. These nanocomposites showcased remarkable electrocatalytic activity and were employed to functionalize the gate electrode of solution-gated graphene transistors (SGGTs). The SGGT-based system exhibited an impressive detection limit of 30 nM for H_2_O_2_. Subsequently, the integration of lactate oxidase into the gate electrode facilitated the development of a compact biosensor capable of real-time lactic acid detection, with a limit of detection of 300 nM and a linear range spanning from 3 to 300 μM. In a complex environment, the lactic acid sensor based on SGGTs effectively detected the presence of lactic acid. Finally, the lactic acid sensor accurately measured lactic acid levels in both cancer and normal cell culture media samples.

Utilizing an aptamer, Hao et al. [66] presented a nanosensor based on an electrolyte-gated GFET for the rapid and sensitive detection of the lung cancer biomarker interleukin-6 (IL-6), yielding improved stability. The aptamer, carrying a negative charge, underwent a conformational change upon binding to IL-6, resulting in the modulation of the carrier concentration in graphene and a measurable alteration in the drain current. The smaller size of the aptamer, compared to other receptors such as antibodies, allows for closer proximity between the charged IL-6 and the graphene surface upon affinity binding, thereby enhancing the detection sensitivity. Consistent sensing performance and higher stability, even after extended-time (more than 24 h) storage at 25 °C, were exhibited by the nanosensor in comparison to antibodies. Furthermore, the rapid transduction of the affinity recognition to IL-6 was enabled by the GFET, facilitating detection within less than 10 min with a detection limit as low as 139 fM.

Gao et al. [39] devised an incredibly sensitive biosensor based on a GFET functionalized with poly-L-lysine (PLL) for the detection of breast cancer miRNAs (Figure 7). PLL was strategically applied to the channel surface of the GFET to facilitate the immobilization of DNA probes through the electrostatic force. The findings demonstrate that the PLL-functionalized GFET (PGFET) biosensors enable the detection of (mi)RNAs with an exceptional detection limit of 1 fM and a remarkably fast detection time of 20 min, using only 2 μL of human serum. The PGFET biosensors exhibit an outstanding sensitivity enhancement of over 113% compared to conventional GFET biosensors. The proposed mechanisms for enhancing the performance of the PGFET biosensor were based on a combination of theoretical modeling and experimental results, highlighting the role of PLL in significantly increasing the density of DNA probes within the channel of the PGFET biosensor, facilitated by the strong electrostatic force.

The utilization of nanoparticle-based DNA probes was found to allow for the increased anchoring of RNA molecules to the sensor surface, resulting in enhanced detection sensitivity when compared to conventional DNA probe immobilization on planar surfaces. The use of phosphorodiamidate morpholino oligomers (PMO)-graphene quantum dots (GQDs)-functionalized rGO-FET biosensors for the ultrasensitive detection of exosomal microRNAs was reported by Li et al. [67]. In the process, the rGO-FET sensor was developed, and a polylysine (PLL) film was deposited on the rGO surface. Subsequently, the GQDs-PMO hybrid was produced and covalently attached to the PLL surface for the detection of exosomal microRNAs (miRNAs). An impressively low detection limit of 85 aM was achieved by using this method, and high specificity was exhibited. Furthermore, exosomal miRNAs in plasma samples were successfully detected, and breast cancer samples were distinguished from healthy samples using the FET sensor.

Deng et al. [68] successfully developed a highly sensitive and rapid biosensor using a solution-gated graphene transistor (SGGT) for the quantification and detection of miRNA-21, an important biomarker associated with early prostate cancer. This unamplified, real-time, and cost-effective biosensor demonstrated excellent performance. Thiol-functionalized ssDNA probes were immobilized on the gold gate, allowing for efficient hybridization with the target miRNA-21 molecules and resulting in Dirac voltage shifts in the SGGT transfer curves. The biosensor exhibited an impressive limit of detection (LOD) of 10^−20^ M and a detection linear range spanning from 10^−20^ to 10^−12^ M. The real-time detection of miRNA-21 molecules could be achieved in 5 min, and the biosensor showed the capability to distinguish even a single-base mismatched miRNA-21 molecule. The biosensor operated on a sensing mechanism that depended on the alteration of the transfer curve caused by the hybridization of the ssDNA probe on the gate electrode with the targeted miRNA. Importantly, the biosensor was tested with blood serum samples obtained from patients without the need for RNA extraction or amplification. The results established the biosensor’s ability to effectively differentiate between cancer patients and the control group, exhibiting a higher sensitivity (100%) compared to PSA detection (58.3%). The integration potential and ease of electrical signal measurement make the SGGT sensors suitable for development as portable detection equipment.

The prior discussion of transistor biosensors based on graphene or graphene derivatives for detecting cancer biomarkers is summarized in Table 2.

## 4. Opportunities and Challenges Regarding Future Applications

The use of CNFET biosensors for cancer detection presents both opportunities and challenges. These biosensors offer high sensitivity, allowing them to detect very low concentrations of target molecules that are crucial for cancer detection. They also enable real-time detection, enabling rapid diagnosis and treatment. Additionally, the relatively low cost and simple manufacturing process of carbon nanomaterials makes these biosensors more accessible for wider use. Their portability allows for easy integration into portable devices, facilitating point-of-care testing in resource-limited or remote areas. Furthermore, their specificity reduces the risk of false positives, improving accuracy in cancer detection. One of the significant advantages of CNFET biosensors is their ability to detect cancer biomarkers without the need for labeling. Functionalized carbon nanomaterials in FET biosensors selectively bind to cancer biomarkers, inducing changes in electrical conductivity or threshold voltage upon interaction. This label-free detection method offers distinct advantages over labeled detection methods [69]. One key advantage is the elimination of the need for additional labeling molecules, such as fluorescent or radioactive tags, which simplifies experimental procedures and reduces costs. Label-free methods directly detect the target molecules without any modifications, providing a more natural and accurate representation of the biological system. Moreover, label-free detection enables the real-time monitoring of biomolecular interactions, allowing for dynamic measurements of binding kinetics and affinity. This real-time analysis provides valuable insights into the temporal aspects of molecular interactions [70]. Additionally, label-free methods offer higher sensitivity and specificity by directly interrogating the binding events, minimizing non-specific interactions and reducing false-positive results. These advantages make label-free detection methods highly desirable in various applications, including biomedical research, diagnostics, and drug discovery, where a comprehensive understanding of molecular interactions is crucial for advancing scientific knowledge and improving patient outcomes.

However, there are challenges that need to be addressed. Carefully integrating CNFET biosensors into the current diagnostic methods is essential to ensure their clinical efficacy. These biosensors can be sensitive to environmental factors like temperature, humidity, and pH, which may affect their reliability and accuracy. Humidity, for instance, can influence the adsorption and desorption of water molecules on the surface of CNTs [71]. The presence of water molecules can introduce additional charges and alter the electrical conductivity of the CNTs. High humidity levels can lead to increased water adsorption, resulting in changes in the electrical response of CNT-based sensors. Therefore, humidity control is crucial to ensure the stability and reproducibility of CNT-based sensing devices. Temperature also plays a significant role in CNT-based sensing [72]. Changes in temperature can affect the electrical conductivity of CNTs due to thermal expansion and phonon scattering effects. Higher temperatures can lead to increased thermal vibrations, which can disrupt the electron transport properties of CNTs and impact their sensing capabilities. Therefore, temperature control and compensation techniques are necessary to maintain the reliability and accuracy of CNT-based sensors. To mitigate the effects of environmental conditions on CNT-based sensing, various strategies can be employed. For humidity control, encapsulation techniques or the use of protective coatings can be implemented to minimize water adsorption and preserve the electrical properties of CNTs. Temperature compensation methods, such as using reference sensors or calibration algorithms, can be employed to account for temperature-induced changes in CNT-based sensor responses. The reliability and accuracy of the CNFET device in the presence of pH variations can be improved through several measures. First, employing appropriate buffer solutions helps maintain a constant pH environment, minimizing the impact of fluctuations and ensuring consistent measurements. Buffers act as stabilizing agents, preventing abrupt pH changes. Additionally, integrating pH sensing capabilities enables the real-time monitoring of pH variations, allowing for the prompt detection and correction of any deviations through continuous monitoring and calibration. Moreover, including a reference electrode in the CNFET setup establishes a stable reference point for pH measurements. This reference point serves as a baseline to distinguish pH changes caused by analyte interactions from environmental fluctuations. This inclusion enhances the device’s reliability and accuracy in pH-sensitive applications [73].

Standardization of the manufacturing process and protocols is necessary to ensure reproducibility and comparability across different laboratories. Standardizing manufacturing processes and protocols for CNFET biosensors across laboratories can be achieved through achieving a widespread consensus regarding protocols, documentation and guidelines, quality control measures, inter-laboratory collaborations and proficiency testing, and the establishment of international standards and certification. Collaboration, conferences, and publications enable the development of standardized protocols. Comprehensive documentation ensures consistency, while quality control measures maintain reproducibility. Inter-laboratory collaborations and proficiency testing identify discrepancies and refine protocols, promoting comparability. International standards, led by organizations like ISO, further enhance reproducibility, and certification processes verify compliance with established standards. Some ISO standards related to CNTs and graphene are listed below:ISO/TS 80004-13:2017 [74]—Nanotechnologies—Vocabulary—Part 13: Graphene and related two-dimensional (2D) materials. This standard provides a vocabulary and terminology related to graphene and other 2D materials, including their structural features, properties, and characterization methods.ISO/TS 10797:2020 [75]—Nanotechnologies—Measurement of carbon nanotube length by transmission electron microscopy. This technical specification describes a method for measuring the length of individual carbon nanotubes using transmission electron microscopy (TEM).ISO/TS 17200:2013 [76]—Nanotechnologies—Analysis of SWCNTs using transmission electron microscopy. This technical specification provides guidelines for the analysis of SWCNTs using transmission electron microscopy, including sample preparation, imaging, and measurement techniques.

Both the stability and shelf life of CNFET biosensors require careful evaluation to ensure long-term effectiveness and reliability. Furthermore, clinical validation is crucial to establish the clinical utility and safety of these biosensors for widespread use in cancer detection.

Despite their promising potential, the commercialization of CNFET biosensors also poses challenges [77]. Producing large-scale, high-quality carbon nanomaterials remains a key challenge, as existing methods are often not scalable or cost-effective enough for commercial production. Integrating carbon nanomaterial into existing technologies and materials requires the development of new fabrication techniques and compatible interfaces. The broader adoption of CNFET is impeded by the imperative to enhance accuracy and reliability. To facilitate the commercialization of CNFETs as biosensors, numerous pivotal engineering obstacles must be surmounted. These challenges encompass optimizing the device structure, developing precise surface bio-functionalization techniques capable of detecting multiple targets, enhancing reproducibility and affordability in large-scale manufacturing, addressing sensor calibration and streamlining user intervention, establishing standardized performance indicators, conducting extensive clinical testing, and effectively managing device lifespan and baseline drift in complex storage conditions [78]. Achieving consistency and reproducibility regarding device performance, integrating systems, and advancing the development of parallel and integrated biosensor arrays represent substantial ongoing hurdles. Nevertheless, CNFETs offer distinctive advantages over conventional methods, and as these technical impediments are gradually overcome, the emergence of commercially viable prototypes is eagerly anticipated. The future of CNFET biosensors holds tremendous potential for realizing multi-target and multi-functional arrays, seamlessly integrating with microfluidic systems, and catalyzing transformative advancements in disease diagnosis and management.

In order to enhance the commercial viability of carbon nanomaterials and facilitate their large-scale production, several advancements and innovations are necessary. Firstly, the development of efficient and cost-effective scalable synthesis methods is crucial. Techniques such as chemical vapor deposition (CVD) and catalytic growth methods need to be improved to increase production rates and reduce costs. Secondly, implementing stringent quality control measures and establishing standardized protocols are essential to ensure consistent material properties and reliability, thereby making carbon nanomaterials more appealing for commercial applications. Additionally, the adaptation of laboratory-scale production processes to larger manufacturing facilities is necessary through the optimization of equipment, automation, and process control to achieve the efficient and cost-effective production of carbon nanomaterials in larger quantities. Effective post-processing techniques are also vital to purify and functionalize the carbon nanomaterials, removing impurities, enhancing material properties, and tailoring their functionality for specific applications. Lastly, exploring ways to reduce production costs by investigating alternative feedstock materials, optimizing energy consumption during synthesis, and implementing cost-efficient purification and separation methods is crucial. These advancements and innovations will contribute to the wider adoption and commercial success of carbon nanomaterials in various industries.

Additionally, questions regarding the long-term stability and environmental impact of carbon nanomaterials need to be addressed before their widespread adoption in various applications. The toxicity associated with carbon nanotubes CNTs and graphene-based materials has garnered significant attention due to their increasing use in various applications [79]. It is crucial to understand and address these toxicity concerns to ensure the safe utilization of these materials. The unique physicochemical properties of CNTs and graphene, such as their high surface area and potential for cellular uptake, can contribute to their toxic effects. To overcome these challenges, several mechanisms can be employed. Surface functionalization, involving the modification of their surfaces with specific functional groups, can enhance biocompatibility and reduce toxicity [80]. Encapsulation within biocompatible matrices or protective coatings can also prevent direct contact with biological systems and minimize the release of toxic components [81]. Additionally, controlling the size, shape, and concentration of these materials can help mitigate their toxicological impact [82]. Thorough toxicity studies evaluating their effects on cellular viability, inflammation, oxidative stress, genotoxicity, and long-term systemic effects are essential for understanding their toxicity mechanisms and establishing appropriate safety guidelines.

Therefore, further research, evaluations, and clinical tests are necessary to validate the effectiveness and safety of CNT- and graphene-based FET biosensors for cancer detection.

Moreover, in addition to CNTs and graphene, other low-dimensional materials have proved useful in the detection of cancer biomarkers, such as those listed below:Two-dimensional materials: Materials such as molybdenum disulfide (MoS2) [83] and Tungsten disulfide (WS2) [84] have emerged as promising candidates for biosensing applications. These materials possess unique electronic and optical properties that make them suitable for detecting cancer biomarkers. Functionalizing the surface of these materials with specific bioreceptors, such as antibodies or aptamers, allows for the selective capture and detection of target biomolecules.Metal nanoparticles: Metal nanoparticles, particularly gold and silver nanoparticles, have been widely investigated for their sensing capabilities in cancer detection [85]. These nanoparticles can be functionalized with specific ligands or probes that bind to cancer biomarkers, leading to changes in their optical properties. These changes can be detected and quantified, providing a sensitive and specific detection method for cancer-related biomarkers.Quantum dots: Quantum dots (QDs) are semiconductor nanocrystals with unique optical and electrical properties [86]. They have shown potential for cancer biomarker detection due to their high brightness, photostability, and tunable emissions. QDs can be functionalized with biomolecules to selectively bind to specific cancer biomarkers, enabling their detection through fluorescence-based assays.Black phosphorus nanosheets: Similar to graphene, black phosphorus nanosheets [87] possess a two-dimensional structure with excellent electronic and optical properties. They have shown promise in cancer biomarker detection due to their large surface area and high carrier mobility. The functionalization of black phosphorus nanosheets with bioreceptors allows for the specific binding and detection of cancer-related biomarkers.Metal-organic frameworks (MOFs): MOFs are highly porous materials composed of metal ions or clusters coordinated with organic ligands. They offer a unique platform for cancer biomarker detection due to their large surface area and customizable pore structures [88]. MOFs can be functionalized with recognition elements to capture and detect specific biomarkers, providing a sensitive and selective detection method.

These low-dimensional materials offer exciting prospects for the detection of cancer biomarkers, complementing the capabilities of CNTs and graphene. Continued research and developments in this field could lead to the advancement of sensitive and reliable diagnostic tools for the detection and monitoring of early-stage cancer.

## 5. Conclusions

In conclusion, the immense potential of these remarkable materials in revolutionizing the field of diagnostics was discussed by reviewing CNT- and graphene-based transistor biosensors used for cancer detection. Unprecedented opportunities for early cancer detection and personalized medicine are offered by their unique properties, including high sensitivity, selectivity, and label-free detection capabilities. New avenues for the precise and real-time monitoring of cancer biomarkers can be opened through the integration of CNTs and graphene into biosensing platforms, bringing improved patient outcomes and more effective treatment strategies closer to realization. While challenges remain in terms of large-scale production and integration into existing technologies, the promising results and continuous advancements in this field inspire confidence, indicating that these biosensors will play a pivotal role in transforming cancer detection by ushering in a new era of accurate, timely, and accessible diagnostics. With each step forward, a future wherein the ability to detect and combat cancer is enhanced by the power of CNT- and graphene-based transistor biosensors draws closer, ultimately bringing hope to those affected by this devastating disease and potentially saving lives.

## Figures and Tables

**Figure 1 biomolecules-13-01024-f001:**
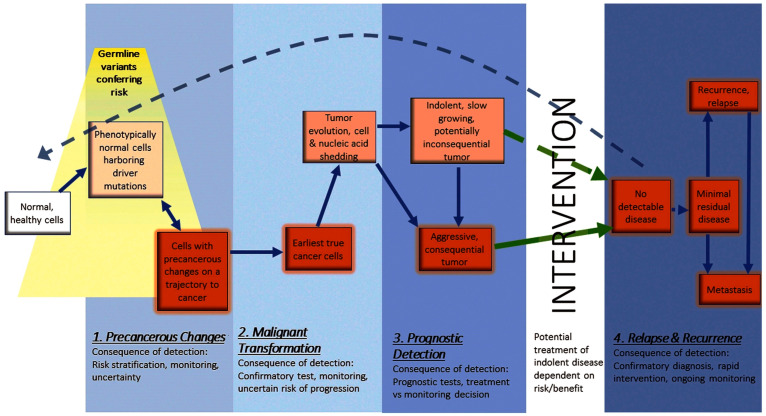
Windows for early detection across the course of cancer progression. (Reproduced with permission from [6]).

**Figure 2 biomolecules-13-01024-f002:**
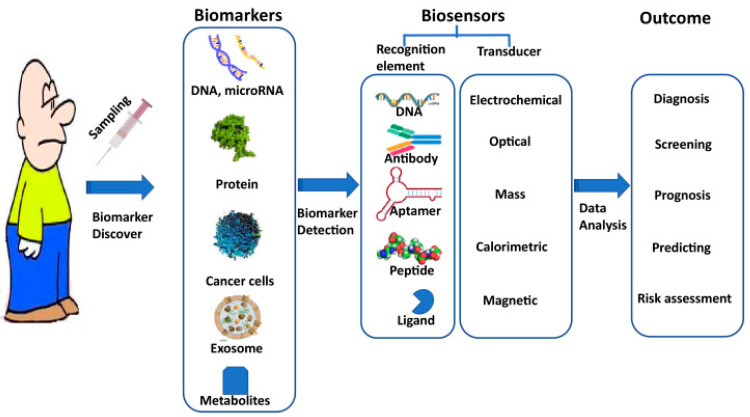
Illustration of the working process of biosensors for cancer diagnosis. (Reproduced with permission from [42]).

**Figure 3 biomolecules-13-01024-f003:**
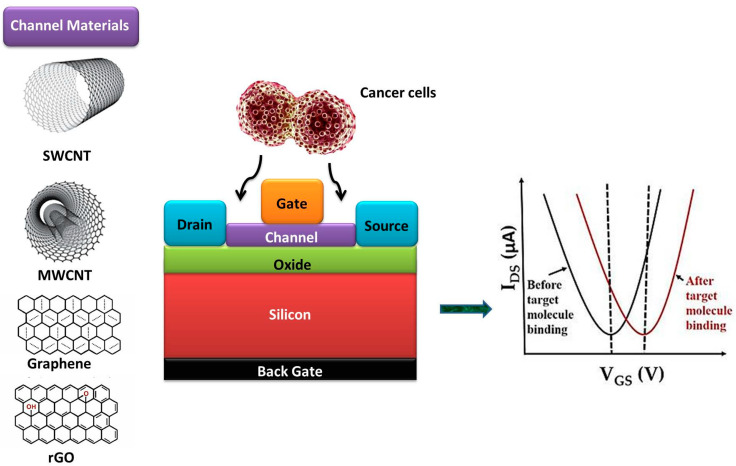
Schematic illustrating the fundamental design of a Field-Effect Transistor (FET) that employs carbon nanomaterials as the channel component to enable the detection of cancer cells by means of electrical measurements.

**Figure 4 biomolecules-13-01024-f004:**
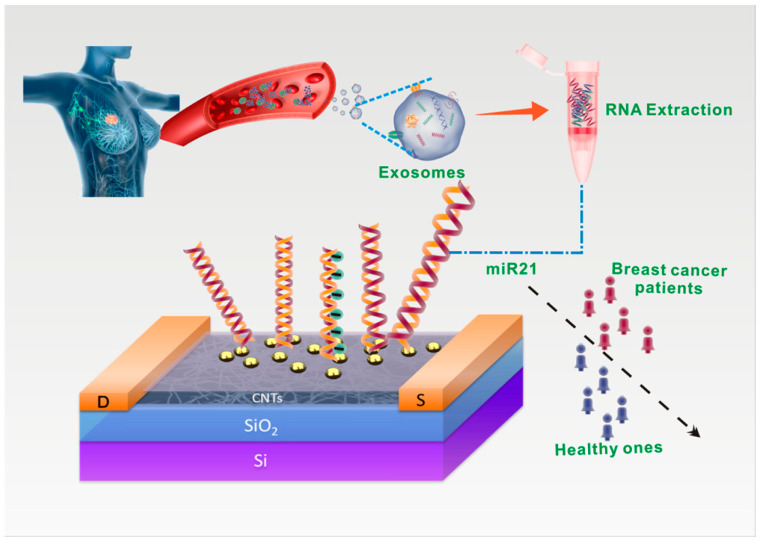
Schematic representation of the highly sensitive detection method for exosomal miR21, employing a biosensor based on a DNA-functionalized carbon nanotube (CNT) field-effect transistor (FET). (Reproduced with permission from [37]).

**Figure 5 biomolecules-13-01024-f005:**
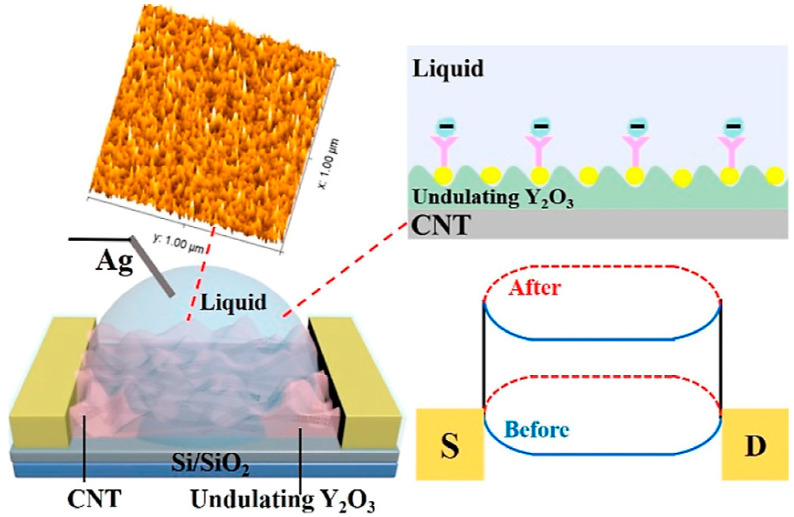
(**Bottom left**) Schematic diagram of the three-dimensional undulating FG CNT-FET biosensor. (**Top left**) AFM images of the morphology of the undulating Y_2_O_3_. (**Right**) Changes in the energy band diagram of sensors with undulating Y_2_O_3_ structures before (blue curve) and after (red curve) the introduction of biomolecules.(Reproduced with permission from [57]).

**Figure 6 biomolecules-13-01024-f006:**
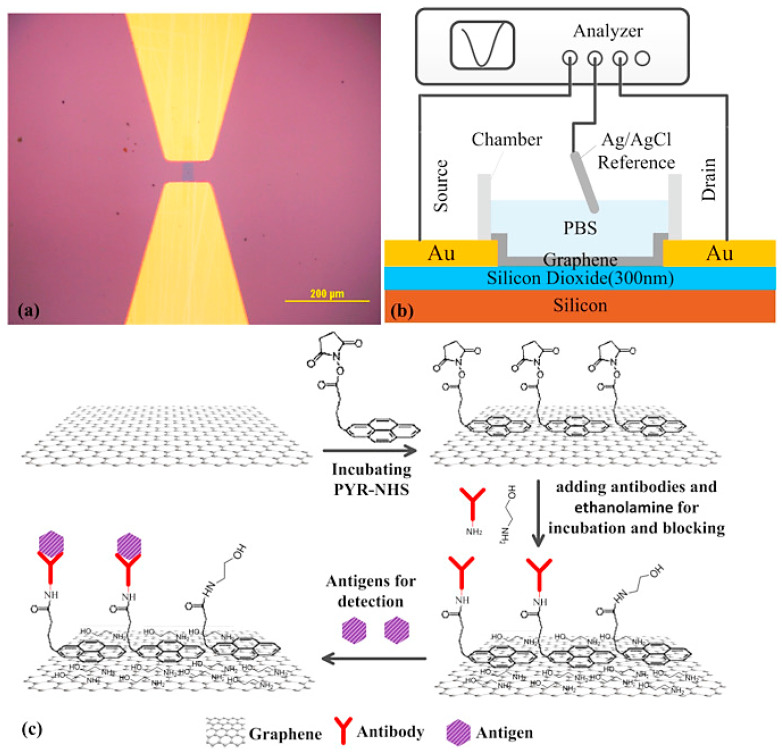
(**a**) Optical micrograph of the graphene channel; (**b**) schematic diagram of solution gated GFET biosensor; (**c**) the schematic diagram of all the modification steps for GFET. (Reproduced with permission from [62]).

**Figure 7 biomolecules-13-01024-f007:**
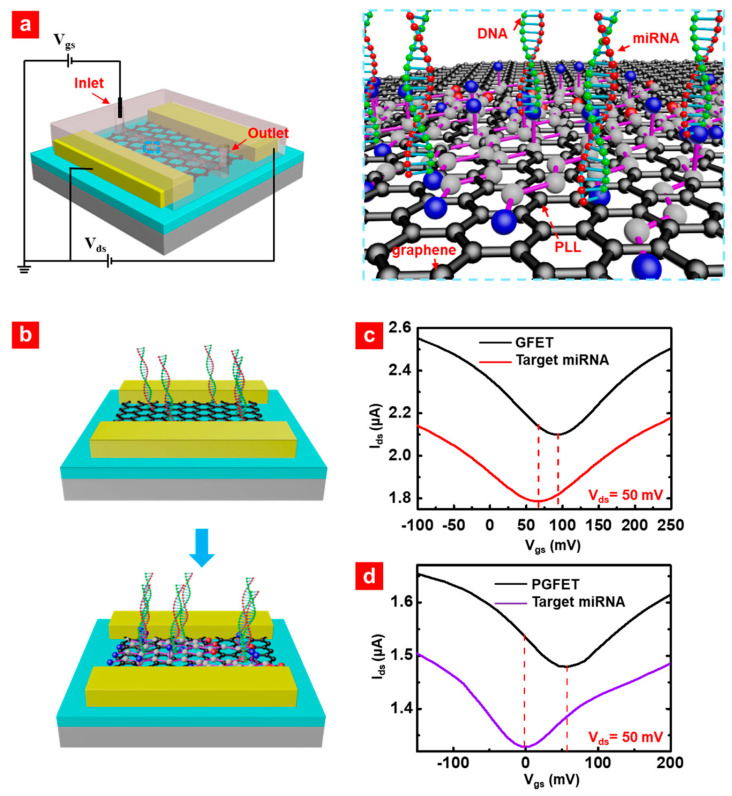
(**a**) PGFET biosensor Schematic. (**b**) Schematic principles of GFET and PGFET for miRNA detection. Comparison of miRNA detection results between GFET (**c**) and PGFET (**d**) biosensors. (Reproduced with permission from [39]).

**Table 1 biomolecules-13-01024-t001:** CNT-based transistor biosensors for the detection of cancer biomarkers.

Substrate Used	Source/ Drain Materials	Gate	Active Layer/ Transducer	Sample Type	Bioreceptors	Target	LOD	Detection Range	Response Time	Disease	Ref
SiO_2_/Si	Ti/Au	Back	CNT	Blood	Insulin-like growth factor 1 receptor (IGF1R) and mouse monoclonal antibody Ab3	Human BT474 and MCF7 breast cancer cells	NA	NA	NA	Breast cancer	Teker et al. [31]
SiO_2_/Si	Au	Solution	CNT	Serum	1-pyrenebutanoic acid succinimidyl ester (PASE)	PSA-ACT complex	1.0 ng/mL	0.1 to 500 ng/mL	NA	Prostate cancer	Kim et al. [32]
SiO_2_/Si	Au	Back	CNT	Serum	Anti-IGF-1 antibody	Insulin-like growth factor-1 (IGF-1)	NA	NA	1 min	Breast Cancer	Jones et al. [51]
SiO_2_/Si	Ti/Pd	Back	CNT		Single chain variable fragment (scFv) protein	osteopontin	1 pg/mL	1 pg/mL to 1 μg/mL	20 min incubation time	Prostate cancer	Lerner et al. [52]
SiO_2_/Si	Au	Solution	CNT	Serum	Carnation Italian ringspot virus p19 protein	MicroRNAs	1 aM	1 aM to 10 fM	60 min incubation time	Cancer	Ramnani et al. [36]
Quartz	Au	Liquid	CNT	Serum	PSA antibody	Prostate-specific antigen	84 pM	up to 1 µM	15 min incubation time	Prostate cancer	Chen et al. [53]
SiO_2_/Si	Au	Back	CNT	Blood	1-Pyrenebutanoic Acid Succinimidyl Ester (PASE) conjugated IL-6 aptamers	Interleukin-6 (IL-6)	1 pg/mL	1 pg/mL to 10 ng/mL	NA	Cancer	Khosravi et al. [54]
SiO_2_/Si	Au	Liquid	CNT	Serum	IL-6R	Interleukin-6 (IL-6)	1.37 pg/mL	1 pg/mL to 100 pg/mL)	NA	Cancer	Chen et al. [55]
SiO_2_/Si	Au	Floating gate	CNT	Plasma	HS-DNA	Exosomal miRNA21	0.87 aM	10 aM-100 nM	NA	Breast Cancer	Li et al. [37]
Al_2_O_3_/Si	CNT	Back	CNT	Blood	Tetrahedral DNA nanostructures	AKT2 gene	2 fM	1 pM to 1 μM	60 min incubation time	Breast cancer	Ma et al. [56]
SiO_2_/Si	Ti/Pd/Au	Floating	CNT	Serum	Monoclonal carcinoembryonic antibody	Carcinoembryonic antigen (CEA)	72 ag/mL	1 fg/mL to 1 ng/mL	NA	Lung cancer	Li et al. [57]

**Table 2 biomolecules-13-01024-t002:** Graphene-based transistor biosensors for the detection of cancer biomarkers.

Substrate Used	Source/ Drain Materials	Gate	Active Layer/ Transducer	Sample Type	Bioreceptors	Target	LOD	Detection Range	Response Time	Disease	Ref
Glass	Ti/Au	Solution	rGO	Serum	PSA monoclonal antibodies	prostate-specific antigen/α1-antichymotrypsin (PSA-ACT)	100 fg/mL	100 fg/mL to 100 ng/mL	10 min	Prostate cancer	Kim et al. [58]
SiO_2_/Si	Au	Solution	Graphene	Serum	Anti-PSA	Prostate-specific antigen	0.4 pg/mL	NA	NA	Prostate cancer	Zhang et al. [59]
SiO_2_/Si	Au	Liquid	rGO	Serum	Peptide nucleic acid (PNA)	miRNA	10 fM	NA	NA	Cancer	Cai et al. [38]
SiO_2_/Si	Au	Back	rGO	Plasma	Polypeptide (JR2EC)	Matrilysin (MMP-7)	10 ng/mL	1 ng/ML to 5 μg/mL	5 min	Cancer	Chen et al. [60]
SiO_2_/Si	Cr/Au	Back	Graphene	Serum	Thiol-containing single-chain variable fragment antibodies (scFv)	HER3	300 fg/mL	300 fg/mL to 300 ng/mL	60 min incubation time	Breast cancer	Rajesh et al. [61]
SiO_2_/Si	Ti/Au	Solution	Graphene	Serum	Anti-CEA	Carcinoembryonic antigen (CEA)	100 pg/ml	Up to 100 ng/mL	NA	Cancer	Zhou et al. [62]
SiO_2_/Si	Cr/Au	Back	Graphene	Blood	Anti-hCG monoclonal antibody	Human Chorionic Gonadotropin (hCG)	1 pg/mL	1 pg/mL to 1 ng/mL	NA	Cancer	Haslam et al. [63]
Poly (methyl methacrylate) (PMMA)	Au	Liquid-ion	rGO	Serum	CA125 ssDNA/MWCNT	Ovarian cancer antigen (CA125)	5.0 × 10^−10^ U/mL	1.0 × 10^−9^ to 1.0 U/mL	40 min incubation time	Ovarian cancer	Majd et al. [33]
Polyethylene terephthalate (PET)	Au	Solution	Graphene	Plasma	Monoclonal anti-alpha-fetoprotein	Alpha-fetoprotein (AFP)	0.1 ng/mL	NA	NA	Hepatocellular carcinoma (Liver Cancer)	Kim et al. [34]
Glass	Au	Solution	Graphene		Probe DNA	TP53 DNA	1 nM	1 nM to 10 μM	NA	Cancer	Kim et al. [64]
Glass	Au	Solution	Graphene	Tumor cells	Nanoceria-loaded porous carbon	Lactic acid	300 nM	3 μM to 300 μM	NA	Cancer	Bi et al. [65]
SiO_2_/Si	Cr/Au	Electrolyte	Graphene	PBS buffer	IL-6-specific aptamer	Interleukin-6 (IL-6)	139 fM		10 min	Lung cancer	Hao et al. [23]
SiO_2_/Si	Ti/Au	Solution	Graphene	Serum	DNA probes	Breast cancer miRNA	1 fM	1 fM to 100 pM	20 min	Breast Cancer	Gao et al. [39]
SiO_2_/Si	Au	Solution	rGO	Blood	Phosphorodiamidate morpholino oligomers (PMO)-graphene quantum dots (GQDs)	Exosomal miRNA	85 aM	100 aM to 1 nM	NA	Breast cancer	Li et al. [67]
Glass	Cr/Au	Solution	Graphene	Serum	Single-stranded DNA (ssDNA) probes	Prostate cancer-relevant miRNA-21	10^−20^ M	10^−20^ to 10^−12^ M	5 min	Prostate cancer	Deng et al. [68]

## Data Availability

Not applicable.

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
