# Peer review of "CNT and Graphene-Based Transistor Biosensors for Cancer Detection: A Review"

_biomolecules, 2023, doi:10.3390/biom13071024_

Round 1
Reviewer 1 Report
This review paper explores the application of carbon nanotube (CNT) and graphene-based transistor biosensors for cancer detection. It discusses the importance of early cancer detection, the role of biosensors in detecting cancer biomarkers, and the opportunities and challenges associated with using carbon nanomaterials in the development of these biosensors. Overall, the paper highlights the potential of CNT and graphene-based biosensors for improving cancer detection and emphasizes the need for further research and development to overcome existing challenges and ensure their clinical efficacy.
However, I believe that the paper can be further improved in two aspects. Firstly, the structure of the paper could be enhanced to provide a clearer flow of information and facilitate better understanding for the readers. Secondly, while the paper focuses on the application of CNT and graphene-based biosensors in cancer detection, it would benefit from a more comprehensive view that incorporates the advancements and sensing mechanisms in different scientific branches related to CNT and graphene. This broader perspective would enhance the understanding of the potential of these materials in biosensing applications. You can find some of my points here:
Section 1:
1- It would be useful to include information on specific cancer screening methods or diagnostic tests that are commonly used for early detection.
2- Improve the quality of Fig. 1.
Section 2:
3- Check the styles of line 107 and 108.
4- It would be helpful to provide a brief explanation of the strengths and limitations of each detection technique mentioned.
5- The section mentions that biosensors and liquid biopsies are emerging as promising methods for non-invasive and early detection of cancer. It would be beneficial to briefly explain what liquid biopsies are and how they contribute to early cancer detection.
6- Provide some specific examples of protein-based biosensors and nucleic acid-based biosensors that have been successfully used for early cancer detection.
7- It would be valuable to provide a brief explanation of how microfluidic biosensors work and what makes them advantageous in terms of sensitivity and multiplexed detection.
Section 3:
8- It would be helpful to provide a brief explanation of the advantages of using CNTs in terms of their electrical properties and sensitivity.
9- It would be beneficial to briefly explain the functionalization process of CNTs with specific bioreceptors, such as antibodies or aptamers.
10- It is necessary to give brief explanations about the sensing mechanism of carbon nanotube and graphene. These references can be considered: DOI: 10.1021/acs.chemrev.8b00340, 10.1021/nl060613v and 10.1016/j.compositesb.2017.09.049.
11- Sensing conditions are very important in the investigation of nanomaterial sensors. Please mention the effects of environmental conditions on sensing such as humidity and temperature in carbon nanotubes. These references can be considered: DOI: 10.3390/s19112464 and 10.1109/JSEN.2020.3038647.
Section 3.1:
12- It would be helpful to provide an explanation of how the antibodies are linked to the CNT-FET devices and their role in the detection process.
13- How do CNT-FET biosensors achieve label-free detection of cancer biomarkers, and what advantages does this approach offer compared to labeled detection methods?
14- Have there been any studies or research exploring the potential challenges or limitations associated with the practical implementation of CNT-based biosensors for cancer detection? It would be interesting to understand the current state of research in terms of addressing issues such as scalability, cost-effectiveness, and translation to clinical settings.
Section 3.2:
15- For the suspended GNR ISFET developed by Zhang et al., what are the mechanisms or factors that contribute to its superior sensitivity and detection limit compared to other graphene-based ISFETs? How does the use of suspended GNRs enhance the performance of the biosensor?
Section 4:
16- Considering the sensitivity of carbon nanomaterial-based FET biosensors to environmental factors, what measures can be implemented to mitigate the impact of temperature, humidity, and pH variations on their reliability and accuracy?
17- How can the standardization of manufacturing processes and protocols for carbon nanomaterial-based FET biosensors be achieved to ensure reproducibility and comparability across different laboratories? Are there any ongoing efforts or guidelines in this area?
18- Mention advancements or innovations are needed to overcome the existing limitations and make carbon nanomaterials more commercially viable, In terms of large-scale production.
19- Discuss potential environmental implications associated with the widespread use of carbon nanomaterials, and how can these concerns be addressed to ensure their sustainable and responsible application.
Author Response
Reviewer 1
Section 1:
Comment 1- It would be useful to include information on specific cancer screening methods or diagnostic tests that are commonly used for early detection.
Response: In Section 2 Cancer Detection and Biosensor, information regarding commonly employed cancer screening methods or diagnostic tests for early detection has already been incorporated (X-rays, computed tomography (CT), magnetic resonance imaging (MRI), ultrasound, etc.,). However, in response to the reviewer's suggestion, additional information on specific cancer screening methods (mammography, pap smear, colonoscopy, prostate-specific antigen (PSA) test) has been included.
Comment 2- Improve the quality of Fig. 1.
Response: The quality of Figure 1 has been enhanced to 300dpi.
Section 2:
Comment 3- Check the styles of line 107 and 108.
Response: The styles of line 107 and 108 have been corrected.
Comment 4- It would be helpful to provide a brief explanation of the strengths and limitations of each detection technique mentioned.
Response: A concise overview of the advantages and disadvantages of X-rays, computed tomography (CT), magnetic resonance imaging (MRI), mammography, pap smear, colonoscopy, prostate-specific antigen (PSA) test, ultrasound, polymerase chain reaction (PCR), and next-generation sequencing (NGS) techniques are incorporated in Section 2 Cancer Detection and Biosensor.
Comment 5- The section mentions that biosensors and liquid biopsies are emerging as promising methods for non-invasive and early detection of cancer. It would be beneficial to briefly explain what liquid biopsies are and how they contribute to early cancer detection.
Response: A brief explanation of liquid biopsy methodology and its contribution to early cancer detection is incorporated in Section 2 Cancer Detection and Biosensor.
Comment 6- Provide some specific examples of protein-based biosensors and nucleic acid-based biosensors that have been successfully used for early cancer detection.
Response: Some specific examples of protein-based biosensors and nucleic acid-based biosensors that have been successfully used for early cancer detection are incorporated in Section 2 Cancer Detection and Biosensor.
Comment 7- It would be valuable to provide a brief explanation of how microfluidic biosensors work and what makes them advantageous in terms of sensitivity and multiplexed detection.
Response: A brief explanation of the working principle of microfluidic biosensors and their advantageous in terms of sensitivity and multiplexed detection is incorporated in Section 2 Cancer Detection and Biosensor.
Section 3:
Comment 8- It would be helpful to provide a brief explanation of the advantages of using CNTs in terms of their electrical properties and sensitivity.
Response: A brief explanation of the advantages of using CNTs in terms of their electrical properties and sensitivity is incorporated in Section 3. Carbon Nanomaterials Based Transistor Biosensors for Cancer Detection.
Comment 9- It would be beneficial to briefly explain the functionalization process of CNTs with specific bioreceptors, such as antibodies or aptamers.
Response: A brief explanation of the functionalization process of CNTs with specific bioreceptors, such as antibodies or aptamers is incorporated in Section 3. Carbon Nanomaterials Based Transistor Biosensors for Cancer Detection.
Comment 10- It is necessary to give brief explanations about the sensing mechanism of carbon nanotube and graphene. These references can be considered: DOI: 10.1021/acs.chemrev.8b00340, 10.1021/nl060613v and 10.1016/j.compositesb.2017.09.049.
Response: A brief explanation about the sensing mechanism of carbon nanotube and graphene is incorporated in Section 3. Carbon Nanomaterials Based Transistor Biosensors for Cancer Detection and the references (DOI: 10.1021/acs.chemrev.8b00340, 10.1021/nl060613v and 10.1016/j.compositesb.2017.09.049.) are cited.
Comment 11- Sensing conditions are very important in the investigation of nanomaterial sensors. Please mention the effects of environmental conditions on sensing such as humidity and temperature in carbon nanotubes. These references can be considered: DOI: 10.3390/s19112464 and 10.1109/JSEN.2020.3038647.
Response: The effects of environmental conditions on sensing such as humidity and temperature in carbon nanotubes are discussed in Section 4. Opportunities and Challenges with Future Scope and the references (DOI: 10.3390/s19112464 and 10.1109/JSEN.2020.3038647) are cited.
Section 3.1:
Comment 12- It would be helpful to provide an explanation of how the antibodies are linked to the CNT-FET devices and their role in the detection process.
Response: A brief explanation of how the antibodies are linked to the CNT-FET devices and their role in the detection process is incorporated in Section 3. Carbon Nanomaterials Based Transistor Biosensors for Cancer Detection.
Comment 13- How do CNT-FET biosensors achieve label-free detection of cancer biomarkers, and what advantages does this approach offer compared to labeled detection methods?
Response: The advantages of label-free detection are described in Section 4. Opportunities and Challenges with Future Scope.
Comment 14- Have there been any studies or research exploring the potential challenges or limitations associated with the practical implementation of CNT-based biosensors for cancer detection? It would be interesting to understand the current state of research in terms of addressing issues such as scalability, cost-effectiveness, and translation to clinical settings.
Response: There are some studies [1, 2] exploring the potential challenges or limitations associated with the practical implementation of carbon nanomaterial-based biosensors for diseases diagnosis. The studies indicated that the broader adoption of carbon nanomaterial-based biosensor is impeded by the imperative to enhance accuracy and reliability. To facilitate the commercialization of carbon nanomaterial-based sensors as biosensors, numerous pivotal engineering obstacles must be surmounted. These challenges encompass optimizing the device structure, developing precise surface bio-functionalization techniques capable of detecting multiple targets, enhancing reproducibility and affordability in large-scale manufacturing, addressing sensor calibration and streamlining user intervention, establishing standardized performance indicators, conducting extensive clinical testing, and effectively managing device lifespan and baseline drift in complex storage conditions. Achieving consistency and reproducibility in device performance, integrating systems, and advancing the development of parallel and integrated biosensor arrays represent substantial ongoing hurdles. Nevertheless, carbon nanomaterial-based biosensors offer distinctive advantages over conventional methods, and as these technical impediments are gradually overcome, the emergence of commercially viable prototypes is eagerly anticipated. The future of carbon nanomaterial-based biosensors holds tremendous potential for realizing multi-target and multi-functional arrays, seamlessly integrating with microfluidic systems, and catalyzing transformative advancements in disease diagnosis and management.
The above discussion along with the references’ is incorporated in Section 4. Opportunities and Challenges with Future Scope of the modified manuscript.
- Li, Z., Xiao, M., Jin, C., & Zhang, Z. (2023). Toward the Commercialization of Carbon Nanotube Field Effect Transistor Biosensors. Biosensors, 13(3), 326.
- Chatterjee, N., Manna, K., Mukherjee, N., & Saha, K. D. (2022). Challenges and future prospects and commercial viability of biosensor-based devices for disease diagnosis. Biosensor Based Advanced Cancer Diagnostics, 333-352.
Section 3.2:
Comment 15- For the suspended GNR ISFET developed by Zhang et al., what are the mechanisms or factors that contribute to its superior sensitivity and detection limit compared to other graphene-based ISFETs? How does the use of suspended GNRs enhance the performance of the biosensor?
Response: The two main factors that contribute to the superior sensitivity and detection limit of suspended GNR ISFET developed by Zhang et al. compared to other graphene-based ISFETs are
- Reduced electrical noise: The suspended configuration of GNRs reduces electrical noise interference from the substrate, resulting in improved signal-to-noise ratio. This reduction in noise enhances the sensitivity and detection limit of the biosensor, allowing for more accurate and reliable detection of analytes.
- Enhanced charge carrier mobility: Suspended GNRs offer improved charge carrier mobility compared to other graphene-based structures. The free-standing nature of GNRs minimizes substrate-induced scattering and phonon scattering, resulting in enhanced charge transport properties. This higher carrier mobility contributes to better sensing performance and lower detection limits.
Because of the above mentioned factors the use of suspended GNRs enhances the performance of the biosensor.
The above discussion has been incorporated in relevant portion of the modified manuscript.
Section 4:
Comment 16- Considering the sensitivity of carbon nanomaterial-based FET biosensors to environmental factors, what measures can be implemented to mitigate the impact of temperature, humidity, and pH variations on their reliability and accuracy?
Response: The brief overview of the measures that can be implemented to mitigate the impact of temperature, humidity, and pH variations on the reliability and accuracy of carbon nanomaterial-based FET biosensors is incorporated in Section 4. Opportunities and Challenges with Future Scope of the modified manuscript.
Comment 17- How can the standardization of manufacturing processes and protocols for carbon nanomaterial-based FET biosensors be achieved to ensure reproducibility and comparability across different laboratories? Are there any ongoing efforts or guidelines in this area?
Response: Standardizing manufacturing processes and protocols for carbon nanomaterial-based carbon nanomaterial-based FET biosensors across laboratories can be achieved through consensus on protocols, documentation and guidelines, quality control measures, inter-laboratory collaborations and proficiency testing, and the establishment of international standards and certification. Collaboration, conferences, and publications enable the development of standardized protocols. Comprehensive documentation ensures consistency, while quality control measures maintain reproducibility. Inter-laboratory collaborations and proficiency testing identify discrepancies and refine protocols, promoting comparability. International standards, led by organizations like ISO, further enhance reproducibility, and certification processes verify compliance with established standards. Some ISO standards related to CNTs and graphene are listed here.
- ISO/TS 80004-13:2017 - Nanotechnologies - Vocabulary - Part 13: Graphene and related two-dimensional (2D) materials. This standard provides a vocabulary and terminology related to graphene and other 2D materials, including their structural features, properties, and characterization methods.
- ISO/TS 10797:2020 - Nanotechnologies - Measurement of carbon nanotube length by transmission electron microscopy. This technical specification describes a method for measuring the length of individual carbon nanotubes using transmission electron microscopy (TEM).
- ISO/TS 17200:2013- Nanotechnologies - Analysis of SWCNTs using transmission electron microscopy. This technical specification provides guidelines for the analysis of SWCNTs using transmission electron microscopy, including sample preparation, imaging, and measurement techniques.
The above discussion is incorporated in Section 4. Opportunities and Challenges with Future Scope of the modified manuscript.
Comment 18- Mention advancements or innovations are needed to overcome the existing limitations and make carbon nanomaterials more commercially viable, In terms of large-scale production.
Response: The advancements or innovations that are needed to overcome the existing limitations and make carbon nanomaterials more commercially viable, especially in terms of large-scale production are discussed in Section 4. Opportunities and Challenges with Future Scope of the modified manuscript.
Comment 19- Discuss potential environmental implications associated with the widespread use of carbon nanomaterials, and how can these concerns be addressed to ensure their sustainable and responsible application.
Response: The potential environmental implications associated with the widespread use of carbon nanomaterial’s toxicity and the same is discussed in Section 4. Opportunities and Challenges with Future Scope of the modified manuscript.
Reviewer 2 Report
Please find attached the reviewer comments.

Needs some improvement.
Author Response
Reviewer 2
Comment 1. The motivation behind this work from the nanotechnology perspective should be highlighted as the review discusses more on the CNT and graphene-based transistor biosensors.
Response: The motivation for this paper stems from the urgent need for improved cancer detection methods that can enable early-stage diagnosis and personalized treatment. Conventional diagnostic approaches often face limitations in terms of sensitivity, selectivity, and real-time monitoring. By harnessing the power of CNT and graphene-based transistor biosensors, this paper aims to shed light on their potential to overcome these limitations and propel cancer diagnostics to new heights. The nanotechnology perspective drives the exploration of CNT and graphene-based biosensors as they offer unparalleled advantages for cancer detection. The ability to directly interface with biological molecules and detect minute concentrations of cancer biomarkers with high sensitivity and selectivity is a game-changer in the fight against cancer. By leveraging the unique properties of CNTs and graphene, nanotechnology strives to unlock novel avenues for early detection, accurate diagnosis, and personalized treatment strategies.
The above discussion is incorporated in Section 2. Cancer Detection and Biosensor of the modified manuscript.
Comment 2. The authors should provide a separate section on conclusions.
Response: A separate section on conclusions has been incorporated.
Comment 3. The utility of other low-dimensional materials for detection of cancer biomarkers needs to be incorporated.
Response: The utility of other low-dimensional materials (e.g. molybdenum disulfide (MoS2), Tungsten disulfide (WS2), Metal nanoparticles, Quantum dots, Black phosphorus nanosheet, Metal-organic frameworks (MOFs)) for detection of cancer biomarkers has been incorporated in Section 4. Opportunities and Challenges with Future Scope of the modified manuscript.
Comment 4. The relevance of the early detection should be highlighted using appropriate references: Micromachines 2023, 14(3), 668; Angewandte Chemie, 62(16), e202217932.
Response: The early detection of cancer is a critical factor in improving patient outcomes, and emerging technologies such as Photonic Crystal Enhanced Fluorescence (PCEF) and Photonic Resonator Absorption Microscopy (PRAM) hold great promise in this endeavor. These techniques have been introduced in Section 2. Cancer Detection and Biosensor of the modified manuscript along with the references suggested.
Comment 5. The toxicity associated with the CNT and graphene-based materials should be discussed and the possible mechanisms to overcome the same can be highlighted.
Response: The toxicity associated with the CNT and graphene-based materials has been discussed and the possible mechanisms to overcome the same are highlighted in in Section 4. Opportunities and Challenges with Future Scope of the modified manuscript.
Comment 6. The rationale for the exploration of plasmonic materials in conjugation with low dimensional substrates can be presented with adequate insights.
Response: The rationales for the exploration of plasmonic materials in conjugation with low dimensional substrates have been introduced in Section 2. Cancer Detection and Biosensor of the modified manuscript.
Round 2
Reviewer 2 Report
The authors address reviewer comments.
Minor corrections needed.